# The Comparison of Serum Exosome Protein Profile in Diagnosis of NSCLC Patients

**DOI:** 10.3390/ijms241813669

**Published:** 2023-09-05

**Authors:** Kamila Baran, Joanna Waśko, Jakub Kryczka, Joanna Boncela, Sławomir Jabłoński, Beata Kolesińska, Ewa Brzeziańska-Lasota, Jacek Kordiak

**Affiliations:** 1Department of Biomedicine and Genetics, Medical University of Lodz, 92-213 Lodz, Poland; ewa.brzezianska@umed.lodz.pl; 2Institute of Organic Chemistry, Faculty of Chemistry, Lodz University of Technology, 90-924 Lodz, Poland; joanna.wasko@p.lodz.pl (J.W.); beata.kolesinska@p.lodz.pl (B.K.); 3Institute of Medical Biology, Polish Academy of Sciences, 93-232 Lodz, Poland; jkryczka@cbm.pan.pl (J.K.); jboncela@cbm.pan.pl (J.B.); 4Department of Thoracic, General and Oncological Surgery, Medical University of Lodz, 90-549 Lodz, Poland; slawomir.jablonski@umed.lodz.pl (S.J.); jacek.kordiak@umed.lodz.pl (J.K.)

**Keywords:** non-small lung cancer, liquid biopsy, serum exosome, protein profile, cancer progression biomarker, lymph node metastasis

## Abstract

A thorough study of the exosomal proteomic cargo may enable the identification of proteins that play an important role in cancer development. The aim of this study was to compare the protein profiles of the serum exosomes derived from non-small lung cancer (NSCLC) patients and healthy volunteers (control) using the high-performance liquid chromatography coupled to mass spectrometry (HPLC-MS) method to identify potentially new diagnostic and/or prognostic protein biomarkers. Proteins exclusively identified in NSCLC and control groups were analyzed using several bioinformatic tools and platforms (FunRich, Vesiclepedia, STRING, and TIMER2.0) to find key protein hubs involved in NSCLC progression and the acquisition of metastatic potential. This analysis revealed 150 NSCLC proteins, which are significantly involved in osmoregulation, cell–cell adhesion, cell motility, and differentiation. Among them, 3 proteins: Interleukin-34 (IL-34), HLA class II histocompatibility antigen, DM alpha chain (HLA-DMA), and HLA class II histocompatibility antigen, DO beta chain (HLA-DOB) were shown to be significantly involved in the cancer-associated fibroblasts (CAFs) and tumor-associated macrophages (TAMs) infiltration processes. Additionally, detected proteins were analyzed according to the presence of lymph node metastasis, showing that differences in frequency of detection of protein FAM166B, killer cell immunoglobulin-like receptor 2DL1, and olfactory receptor 52R1 correlate with the N feature according to the TNM Classification of Malignant Tumors. These results prove their involvement in NSCLC lymph node spread and metastasis. However, this study requires further investigation.

## 1. Introduction

Lung cancer is one of the most commonly diagnosed cancers and a leading cause of cancer-related deaths worldwide. In 2020, over 2.2 million new cases were diagnosed and over 1.7 million deaths were registered [1]. Commonly, lung cancer comprises small cell lung cancer (SCLC), accounting for 15–20% of lung cancer cases, and non-small cell lung cancer (NSCLC), accounting for 80–85% of cases. The latter includes other types of lung cancers: Adenocarcinoma, which is the most common cancer type; squamous cell carcinoma; and large cell carcinoma [2,3]. Despite making a lot of efforts to early detect lung cancer, a large proportion of NSCLC patients (47%) are diagnosed at advanced stages—stage III/IV according to the American Joint Committee on Cancer (AJCC) classification—when the tumor has already spread to lymph nodes or to distant organs [4]. The 5-year survival rate for patients diagnosed at an advanced stage is approximately 6.1%, whereas the rate for patients diagnosed at an early stage is 61.4% [4]. Thus, in order to improve the clinical outcome and overall survival rate, it is crucial to develop novel, accurate diagnostic and/or prognostic biomarkers for NSCLC and establish a fast and efficient early detection system.

Liquid biopsy represents a novel and non-invasive approach to testing tumor biomarkers in biological fluids, such as blood, saliva, urine, cerebrospinal fluid, and bronchoalveolar lavage fluid (BALF). It allows for the avoidance of mistaken evaluation of tumors arising from inaccessibility to tumor tissue and its clonal heterogeneity [5]. This method involves an analysis of circulating tumor cells (CTCs) or materials secreted by tumors into the circulation, such as cell-free nucleic acids (cfNAs) and extracellular vesicles (EVs) [6].

Exosomes belong to a group of small EVs (diameter 30 nm–100 nm), carrying a variety of bioactive molecules, such as nucleic acids (RNA as well as DNA molecules), proteins, and lipids [7]. The exosome content is protected from degradation, making it more stable than proteins and nucleic acids in body fluids [8,9]. Due to the fact that exosomes harbor plenty of biological information that may be transferred between cells, they play multiple roles in physiological and pathological conditions. Tumor-derived exosomes participate in the development of metastasis in lung cancer by enhancing cancer cell proliferation and migration, promoting angiogenesis and epithelial–mesenchymal transformation (EMT), and also suppressing immunity [10,11,12].

Changes in the protein profile in serum exosomes from NSCLC patients were found to be potentially useful in the diagnosis of this cancer [13,14,15,16,17,18]. However, there are few studies on the correlation of exosomal protein patterns with NSCLC progression and the spread of the tumor in the lymphatic system [19,20]. Additionally, precise information is needed regarding the function of exosomal proteins in the development of lung cancer.

This study aims to evaluate changes in the serum exosome protein profile of NSCLC patients. We analyzed differences in the levels of exosome proteins in NSCLC patients to identify a potential prognostic biomarker of its metastasis. We also performed an enrichment analysis of exosomal proteins detected exclusively in NSCLC patients and controls to identify key protein hubs involved in the acquisition of metastatic potential.

## 2. Results

### 2.1. LC/MS Analysis of Protein Profiles of Serum Exosomes of NSCLC Patients vs. Controls

To establish potential biomarkers that could be utilized during liquid biopsy to detect the presence of lung cancer, a proteomic analysis that assessed the protein content of the exosomes isolated from the serum of 15 NSCLC patients and 16 healthy donors (control group) was performed. Our study identified 150 proteins exclusive to the exosomes from the NSCLC group (Appendix A) and 86 proteins exclusive to the control group (Appendix A).

Olfactory receptor 9G9 and olfactory receptor 2AT4T were the most common proteins found only in the exosomes of NSCLC patients. They were detected in 80% of the samples (Appendix A). In 73% of patients with NSCLC, the homeobox proteins DBX1 and R-spondin-2 were detected. Olfactory receptor 52R1 and neurexophilin-3 were identified in 67% of samples. Killer cell immunoglobulin-like receptor 2DL1, protein FAM166B, olfactory receptor 13C4, and gamma-interferon-inducible lysosomal thiol reductase were present in 60% of NSCLC serum exosomes. Olfactory receptor 2T4, myeloid leukemia factor 2, pulmonary surfactant-associated protein A2, transcription factor JunB, thioredoxin-dependent peroxide reductase, inactive phospholipid phosphatase 7, and odd-skipped-related 1 protein were found in 53% of samples obtained from NSCLC patients. Another 133 detected proteins were present in less than 50% of NSCLC patients.

In the control group, olfactory receptor 14I1, which was found most frequently, was detected in 81% of samples (Appendix A). Putative olfactory receptor 52L2 was identified in 75% of samples obtained from the control group, while aquaporin-12A, olfactory receptor 8B8, and olfactory receptor 52I1 were found in 44% of healthy donors. Another 81 proteins, identified by proteomic analyses, were less common in the serum exosomes of the control group.

### 2.2. Analysis of Hierarchical Clustering

The NSCLC and control patient groups present noticeable heterogeneity regarding the frequency of occurrence of their respective exosome-derived proteins. Thus, to analyze the possible discriminative ability of isolated exosomal proteins, hierarchical clustering analysis (via Orange data mining software, version 3.31.3) was performed using the percentage sheer of the chosen top 40 proteins exclusively observed in the respective group (Figure 1). The goal of this analysis is to build a bidirectional tree diagram (dendrogram), where the subjects (proteins on one dendrogram and patient groups on the other) that were viewed as most similar are placed on branches that are close together [21]. The performed analysis clearly proves the differences between NSCLC and control groups, suggesting the potential discriminating abilities of selected proteins and their eventual usefulness as predictive markers.

### 2.3. Analysis of Protein–Protein Interactions in Serum Exosome-Derived Proteins

Proteins isolated from serum exosomes of NSCLC patients and the control group (healthy patients) were analyzed using the STRING version 11.0 online platform. PPI networks for 150 proteins exclusively found in NSCLC and 86 proteins exclusively found in the control group (Figure 2) were visualized with respective KEGG functional enrichments. A KEGG pathways analysis showed that exosome proteins exclusive to the NSCLC group, with high statistical significance, are involved in MHC I and MHC II antigen processing and presentation (hsa04612—“Antigen processing and presentation—Homo sapiens”) and Graft-versus-host disease (hsa05332—“Graft-versus-host disease—Homo sapiens”), whereas control group-specific proteins were enriched in olfactory signal detection, processing, and transduction (hsa04740—“Olfactory transduction—Homo sapiens”).

Next, an analysis of fold change protein enrichment in molecular functions and biological processes was performed using FunRich software (v3.1.3), supported by the Vesiclepedia database, which provides the best insight into extracellular vesicle-derived data (Figure 3). A Venn diagram proved that the chosen and analyzed exosome proteins from NSCLC patients and the control group are specific for each group, as they present non-shared proteins. Additionally, proteins identified in both groups were compared with the total number of exosome proteins from lung cancer reported on the Vesiclepedia database. 109 proteins present in the NSCLC group and 63 from the control group were not previously documented to be present in exosomes of NSCLC patients in the Vesiclopedia database (data valid on 19 June 2023). The highest fold change enrichment in molecular functions (in comparison to background) for exosome proteins exclusively identified in the NSCLC group was observed in the activity of RNA methyltransferase, MHC I and II receptor, serine-type peptidase, and peroxidase. Furthermore, osmoregulation was by far the most enriched biological process for the NSCLC group, with cell–cell adhesion, cell motility, and cell differentiation also presenting a significantly enhanced fold change. On the other hand, the highest fold change enrichment in molecular functions for exosome proteins exclusively found in the control group includes cytokine binding and the activity of B-cell receptors, water channels, complement, and acetyltransferase. This enrichment corresponds well to the enrichment in biological processes regarding complement activation, cytoskeleton organization, cell growth, and regulation of proliferation.

Following, a comparative analysis of protein functional enrichment in the cellular component, molecular functions, biological pathways, and biological processes was performed using FunRich software (v3.1.3), supported by the Vesiclepedia database that provides the best insight into extracellular vesicles-derived data (Figure 4). The obtained data highlighted significant differences between the two compared groups. Regarding cellular components, exosome proteins exclusively found in the NSCLC patient group are prevalently involved in molecular functions in the nucleus, cytoplasm, lysosome, and extracellular space, such as methyltransferase activity, MHC class I and II receptor activity, and regulation of transcription factor activity, whereas control group proteins are mainly integral to the plasma membrane and nuclei and involved in the activity of G-protein receptors. The main biological pathways and processes of the NSCLC group include cell motility, regulation of DNA replication, p53-independent DNA damage response, and mesenchymal-to-epithelial transition (MET). Additionally, a large number of biological processes in which the selected proteins are involved are currently unknown. Control group proteins are enriched in various signal transduction pathways (mainly via GPCR), cell communication, apoptosis, and epithelial-to-mesenchymal transition (EMT).

### 2.4. Analysis of Serum Exosome-Derived Protein Hub Clusters

The STRING database and Markov clustering (MCL) algorithm were utilized to analyze major clusters of protein–protein interactions in both groups of exclusively identified exosome proteins. Eight clusters from the NSCLC and control groups were chosen and presented in Figure 5.

A detailed enrichment analysis of each cluster, presented in Appendix A, highlighted proteins whose cooperation is significantly involved in given processes. In the control group, the most prominent cluster hubs and their respective processes are CD81 PLAUR CDC42—a cluster mainly involved in positive regulation of cell–cell adhesion, CD48 MLST8 RAC2—natural killer cell-mediated cytotoxicity, SSR2 SPCS2—protein export, UBE2E2 UBE2J1—ubiquitin-mediated proteolysis, IL20RB PDGFB—JAK-STAT signaling pathway and finally, MEAF6 ING4—a cluster involved in histone acetylation. Clusters formed by proteins exclusive to NSCLC exosomes demonstrate more complex enrichments. BATF CD40LG IL34 LYL1 SPI1 cluster is predominantly involved in hematopoietic stem cell differentiation, HLA-DMA HLA-DOB IFI30 RFXANK enriches MHC class II antigen presentation, Th1 and Th2 cell differentiation, Th17 cell differentiation, cell adhesion molecules that correspond with the GZMA KIR2DL1 KLRC1 cluster, being involved in MHC class I protein complex binding, the ACRV1 LYPD3 SPACA3 cluster, and the LYPD5 LYPD8 cluster, responsible for the synthesis of GPI-anchored proteins and u-PAR/Ly-6 domain maturation, and DKK2 RSPO2 WNT3A, involved in positive regulation of the canonical WNT signaling pathway. Additionally, cluster SPSB4 KCTD6 COMMD5 is involved in neddylation, CLDN12 CLDN15 cluster is involved in calcium-independent cell–cell adhesion via plasma membrane cell-adhesion molecules, tight junction interactions, and epithelial-to-mesenchymal transition in cancer, and CTSB M6PR is involved in lysosome activity.

### 2.5. Analysis of the Impact of NSCLC Patients Serum Exosome-Derived Proteins on Immune Cell Infiltration

Exosomes, secreted by cancer cells, are often responsible for creating an environment that facilitates tumor angiogenesis, tumor progression, invasion, and metastasis [22]. Thus, the involvement of exclusive exosome proteins from the NSCLC group in tumor infiltration was analyzed using the Tumor Immune Estimation Resource—TIMER2.0 (http://timer.cistrome.org/, accessed on 10 August 2023) platform and the xCell algorithm. Three identified proteins—IL34, HLA-DMA, and HLA-DOB—were found to be involved in both CAFs and M2 macrophage infiltration (Figure 6), presenting significant correlation efficiency Rho for respective M2 macrophage and CAF infiltration levels: 0.187 and 0.429 for IL34, 0.333 and 0.468 for HLA-DMA, and 0.343 and 0.217 for HLA-DOB. CAFs and M2 macrophages are the main components of the TME that promote tumor progression [23].

After establishing the major differences between NSCLC and control groups that revealed potential biomarkers that could be utilized during liquid biopsy to detect lung cancer and predict TME interaction, our focus was shifted to identifying potential discriminative features that could predict lung cancer type and stage (TNM) in a non-invasive manner.

### 2.6. Protein Profiles of Serum Exosomes of NSCLC Patients According to the Histopathological Subtype of Tumor

Among proteins identified only in serum exosomes of NSCLC patients, HUWE1-associated protein modifying stress responses, mitochondrial adenylate kinase 2, cytochrome c oxidase assembly protein COX11, mitochondrial homeobox protein Nkx-2.8, insulin-like growth factor-binding protein 1, kallikrein-8, neuroblastoma suppressor of tumorigenicity 1, potassium channel tetramerization domain containing 6, and zinc finger FYVE domain-containing protein 21 were detected only in patients with diagnosed AC subtypes of NSCLC (Appendix A).

Moreover, COMM domain-containing protein 5, lactosylceramide 4-alpha-galactosyltransferase, sperm acrosome membrane-associated protein 3, TCF3 fusion partner, and vascular endothelial growth factor D were found only in patients with diagnosed SCC subtypes (Figure 7, Appendix A).

### 2.7. Protein Profiles of Serum Exosomes of NSCLC Patients According to the Tumor Size

In NSCLC patients with T1 tumor, olfactory receptor 2AT4, and homeobox protein DBX1 were observed in all samples (100%); however, these proteins were also detected in 80% and 60% of the samples of patients with T2 tumors, respectively, and were also observed in 50% of the samples of patients with T3 + T4 tumors (Appendix A). Cation-dependent mannose-6-phosphate receptor was found only in the group of patients with the T1 tumor (67% of samples).

The olfactory receptor 9G9 was detected in 83% and 100% of the samples of patients with the T1 and T2 tumors, respectively, but in 50% of the samples of patients with the T3 + T4 tumors (Appendix A). In 83% of T1 tumor samples, R-spondin-2, killer cell immunoglobulin-like receptor 2DL1, olfactory receptor 52R1, olfactory receptor 13C4, and inactive phospholipid phosphatase 7 were observed.

Moreover, R-spondin-2 and olfactory receptor 52R1 were also detected in 80% of T2 tumor samples, but R-spondin-2 was also found in 50% of the samples of patients with T3 + T4 tumors, and olfactory receptor 52R1 was found in 25% of the samples of patients in this group (Appendix A).

Killer cell immunoglobulin-like receptor 2DL1, olfactory receptor 13C4, and inactive phospholipid phosphatase 7 were also found in 60% of the T2 tumor samples. In patients with the T3 + T4 tumor, killer cell immunoglobulin-like receptor 2DL1 was observed in 50% of the samples, and olfactory receptor 13C4 was detected in 25% of samples; however, inactive phospholipid phosphatase 7 was not observed in this group of patients (Appendix A).

In NSCLC patients with T3 + T4 tumors, 19 proteins were not detected; however, they were found in 60% and 67% of the samples of the patients with T2 and T1 tumors, respectively (Appendix A). Pleckstrin homology domain-containing family F member 1 and protein lyl-1 were also not detected in NSCLC patients with the T3 + T4 tumors; however, they were observed in 40% and 67% of the samples of patients with the T2 and T1 tumors, respectively.

### 2.8. Protein Profiles of Serum Exosomes of NSCLC Patients According to the State of Regional Lymph Nodes

In NSCLC patients without metastasis to lymph nodes (N0 feature), olfactory receptor 2AT4 and R-spondin-2 were detected in all samples (100%); however, these proteins were also observed in 67% and 50% of the samples of patients with the N1 feature, respectively, and were also detected in 75% of the samples of patients with the N2 feature (Appendix A). In 80% of NSCLC patients with N0, the following proteins were detected: Olfactory receptor 9G9, homeobox protein DBX1, olfactory receptor 52R1, killer cell immunoglobulin-like receptor 2DL1, gamma-interferon-inducible lysosomal thiol reductase, and neurexophilin-3 were detected. Olfactory receptor 9G9 was also found in 83% of NSCLC patients with N1 features and 75% of patients with N2 features. Homeobox protein DBX1, olfactory receptor 52R1, and killer cell immunoglobulin-like receptor 2DL1 were identified in 67% of patients with N1 features and in 75%, 50%, and 25% of patients with N2 features, respectively. Gamma-interferon-inducible lysosomal thiol reductase and neurexophilin-3 were detected in 33% of NSCLC patients with N1 features. In 75% of NSCLC patients with N2 features, gamma-interferon-inducible lysosomal thiol reductase was identified, while neurexophilin-3 was found in all of them.

Protein FAM166B was detected in all samples (100%) of NSCLC patients with N2 features in 50% of patients with N1 features, and in 40% of patients with N0 features (Appendix A). Receptor expression-enhancing protein 2, GTP cyclohydrolase 1, and Cdc42 effector protein 3 were detected in 75% of NSCLC patients with N2 features, but only in 17% of patients with N1 features and 20% of patients with N0 features. In 50% of patients with N2 features, cytochrome c oxidase assembly protein COX11, corrinoid adenosyltransferase MMAB, and putative cytosolic acyl coenzyme A thioester hydrolase-like were detected; however, they were not found in patients with N1 and N0 features. Additionally, vascular endothelial growth factor D, lactosylceramide 4-alpha-galactosyltransferase, ARL14 effector protein, ER membrane protein complex subunit, CD40 ligand, Ras-like protein family member 11A, protein YIPF4, tetraspanin-3, uncharacterized protein C22orf46, proteasome subunit alpha type-6, and mannose-P-dolichol utilization defect 1 protein were not found in any of the patients with lymph node metastases, but these proteins were detected in 40% of non-metastatic patients.

To analyze the heterogeneity and eventual similarities inside different NSCLC subgroups, N0, N1, and N2 feature hierarchical clustering analysis (via Orange data mining software) was performed analogously to the previous section using 16 proteins that were observed in at least 50% of total NSCLC (Figure 8). N0 and N1 features are located on the same arm of the dendrogram, proving that they present the highest degree of similarity compared to N2 regarding the frequency of protein occurrence inside respective NSCLC subgroups. Importantly, proteins FAM166B, olfactory receptor 52R1, and killer cell immunoglobulin-like receptor 2DL1 present the highest discriminating abilities, thus potentially being utilized to distinguish N features during liquid biopsy.

## 3. Discussion

Until now, tissue biopsy has represented the gold standard in the diagnosis of lung cancer [24]. In everyday practice, the availability of the tumor and the clinical condition of patients do not often allow for sufficient quantity and quality of tumor sampling [25,26]. Liquid biopsy is a minimally invasive and easily repeatable method of sampling, isolating, and testing tumor-derived analytes from biological fluids [27]. To date, ctDNA is the only circulating biomarker approved for the selection of NSCLC patients for targeted therapy [28]. Other tumor-derived materials analyzed in liquid biopsy, such as circulating tumor cells (CTC), circulating tumor RNA (ctRNA), and tumor extracellular vesicles (EV), are still under clinical trials [28].

Recently, the biological significance of exomes in the development of cancer, including lung cancer, and the clinical utility of exosomes have been extensively studied [8,12,29,30]. Cancer cells have been shown to secrete approximately ten times as many exosomes as normal cells, and these tumor-derived exosomes facilitate cellular communication by delivering growth factors, chemokines, RNA, proteins, and lipids [31,32]. The discovery of the exact cargo of exosomes would allow us to better understand the development of cancer metastasis and even provide strategies to identify metastatic lesions.

Our proteomic analysis of the serum exosome protein profile of NSCLC patients revealed a great variety of proteins transported by exosomes in this group. The most common proteins identified in 80% of the serum exosomes of NSCLC patients included olfactory receptors 9G9 and 2AT4. Homeobox proteins DBX1 and R-spondin-2 were also often detected in NSCLC patients (73%). All these proteins were not found in the control group, and our results suggest that they may be used as new potential diagnostic biomarkers for NSCLC patients in liquid biopsy. However, this observation requires further investigation. On the other hand, we observed that another protein that belongs to the olfactory receptor family—olfactory receptor 14I1—was the most frequent protein detected only in serum exosomes of the control group. In addition to the above-mentioned receptors, another 14 proteins belonging to the olfactory receptor family were only detected in the control group, and six more olfactory receptors were exclusively detected in NSCLC patients. To our knowledge, our study is the first to report the presence of many olfactory receptors in the serum exosomes of NSCLC patients. Olfactory receptors (ORs) represent one of the largest multi-gene families in the human genome, consisting of approximately 900 genes and pseudogenes (with 400 functional genes) [33]. They are G protein-coupled receptors (GPCRs) that are predominantly found in the olfactory epithelium. They are essential for detecting and distinguishing among odorants; however, many ORs are ectopically expressed in other tissues and involved in cancer development, including lung cancer [34,35]. A study by Giandomenico et al. revealed that olfactory receptor 51E1 (OR51E1) was highly expressed mainly in the tumor cell membrane of both primary lung carcinoids (typical and atypical carcinoids) and of regional/distant metastases in comparison with normal lung tissue [34]. Activation of ORs has been revealed to regulate many signal pathways involved in the survival and proliferation of cells, and some ORs demonstrate high identity with chemokine receptors and play a role in cell migration [33,35]. An in vivo study conducted by Vadevoo et al. revealed that the activation of OR51E2 in the membrane of macrophages induces M2 polarization and leads to the generation of tumor-associated macrophages (TAMs), which promote lung tumor progression and maintain the tumor microenvironment (TME) immunosuppressive [36]. However, a study by Kalbe et al. demonstrated that the activation of OR2J3 in cells of the NSCLC cell line A549 inhibits tumor cell proliferation and migration and induces apoptosis via activation of phoshoinositol-3 kinase (PI3K) [35]. The above studies demonstrate the different roles of various receptors belonging to the OR family in lung tumor development and progression.

Homeobox protein DBX1, also known as developing brain homeobox protein 1, is important in spinal cord development [37]. However, little is known about the biological function of DBX1 in lung cancer. Our study identified other proteins belonging to the homeobox family in serum exosomes of NSCLC patients, such as homeobox protein BarH-like 1, homeobox protein Hox-C4, homeobox protein MOX-1, homeobox protein Nkx-2.8, and homeobox protein notochord. Nevertheless, they were detected in a smaller number of study samples. Among these proteins, the role of the homeobox protein BarH-like 1 and the homeobox protein Nkx-2.8 in lung cancer development has been best confirmed so far [38,39,40]. Chen et al. revealed that the homeobox protein BarH-like 1 demonstrates suppressive activity and that its expression is downregulated in NSCLC tissues [38]. Downregulation of Barx2 expression enhances cell proliferation, migration, and aerobic glycolysis by promoting the Wnt/β-catenin signaling pathway. Furthermore, the authors demonstrated that the expression level of this protein shows a negative correlation with the stage of cancer development (AJCC staging), and survival analysis reveals that Barx2 expression level is an independent prognostic factor for NSCLC patients. An in vitro study by Kendall et al. revealed that the homeobox protein Nkx-2.8 is characterized by oncogenic activity. The above and thyroid transcription factor-1 functionally cooperate in promoting proliferation of lung cancer cell growth and enhancing tumorigenicity [39]. Furthermore, Hsu et al. demonstrated that lung cancer cell lines, showing coactivation of the TTF-1 and NKX2–8 pathways, exhibit resistance to cisplatin, which is a standard of care in the treatment of NSCLC [40].

In our analysis, we also found that exosome proteins specific for NSCLC are significantly involved in the pathway responsible for MHC I and MHC II antigen processing and presentation and graft-versus-host disease, while most exosome proteins exclusive to the control group are involved in olfactory signal detection, processing, and transduction. It has been shown that exosomes secreted by different professional antigen-presenting cells (APCs) play crucial roles in carrying and presenting functional MHC-peptide complexes and modulating antigen-specific T-cell activation. However, expression of the MHC II complex is not restricted only to APCs and has been proven to be present in lung cancer cells. This fact enables cancer cells to directly present tumor neoantigens to CD4+ T cells, providing their activation in the tumor microenvironment (TME) in an APC-independent manner [41]. Recent data demonstrate that the tumor-expressed MHC II complex is associated with favorable outcomes of anti-tumor immunotherapies [42]. CD4+ T-cell activation (via tumor-specific MHC II) leads to increased interferon-gamma (IFN-γ) secretion, thus enhancing CXCL9 production and PD-L1 expression, culminating in an immunotherapy-sensitive tumor phenotype [41]. Our data clearly prove that analysis of NSCLC exosome-derived proteins may be used to predict immunotherapy outcomes and thus may serve as a prognostic factor in future personalized therapy. Interestingly, a proteomic analysis of tumor-derived exosomes (TEXs) cargo revealed that they contain MHC molecules as well as the pro-apoptotic member of the TNF family—Fas ligand (FasL), which induces apoptosis of T cells and thus has anti-inflammatory effects [43].

Our study revealed the presence of interleukin-34 (IL-34), HLA class II histocompatibility antigen, DM alpha chain (HLA-DMA), and HLA class II histocompatibility antigen, DO beta chain (HLA-DOB) in serum exosomes of NSCLC patients, which, according to TIMER2.0 analysis, have a significant impact on tumor infiltration by CAFs and M2 macrophages contributing to tumor progression. Moreover, it was revealed that CAF infiltration also contributes to the successful treatment of lung cancer patients due to their role in acquiring resistance to epidermal growth factor receptor tyrosine kinase inhibitors (EGFR) [44]. Cancer-associated fibroblasts (CAFs) account for a major component of tumor stroma, and their activation by paracrine factors can promote tumor growth, angiogenesis, invasion, and metastasis, as well as extracellular matrix (ECM) remodeling [45]. Studies have revealed the mutual effects of CAFs and the tumor immune microenvironment (TIME) as a key factor in promoting tumor progression [46,47]. Tumor-associated macrophages (TAMs) are immune cells that migrate from the blood to the tumor site in response to the action of chemotactic factors produced in the cancer microenvironment [48,49,50]. TAMs are classified into classically activated, pro-inflammatory M1 macrophages and alternatively activated, anti-inflammatory M2 macrophages. M1 macrophages exert mainly an anti-tumor function by mediating antibody-dependent cellular cytotoxicity and producing reactive oxygen species (ROS) and tumor necrosis factor (TNF), while M2 macrophages contribute to malignancy through the production of tumor and angiogenic growth factors, extracellular matrix remodeling, and immunosuppression. Additionally, high numbers of these cells often correlate with a bad prognosis and therapeutic resistance [51,52].

Among the exosome proteins detected in the NSCLC group, Dickkopf-related protein 2, R-spondin-2, and protein Wnt-3a are vital regulators of the canonical Wnt signaling pathway. Wnt3a is a classical ligand in the canonical Wnt signaling pathway and binds to Frizzled (Fz) receptors and low-density lipoprotein receptor-related protein 5/6 (LRP5/6) and induces accumulation of β-catenin, while R-spondin-2 prevents degradation and membrane clearance of FZD and LRP5/6, resulting in sensitization of cells to Wnt ligands [53]. As a result of the activation of receptors, β-catenin translocates to the nucleus, where it induces the expression of canonical Wnt target genes, such as CCND1 (Cyclin D1), MYC (c-Myc), and CD44 [54,55]. Wnt signaling plays a pivotal role in the development of lung cancer through the regulation of cell proliferation, migration, and apoptosis. It is also connected to EMT by the upregulation of EMT regulators such as Snail2 and ZEB1 [56]. Interestingly, Dickkopf-related protein 2 is an LRP5/6 antagonist that prevents the formation of the Fz-LRP5/6 complex and contributes to silencing the Wnt signaling pathway. However, it can play a role as an oncogene [55]. Its association with tumor immune evasion in some subsets of melanoma and colorectal tumors has also been reported, where DKK-2 depletion activates natural killer (NK) cells and CD8+ T cells and inhibits tumor progression [57].

Moreover, our analysis revealed a significant role for exosome proteins from NSCLC patients in osmoregulation as well as cell–cell adhesion, cell motility, and differentiation, while exosome proteins derived from controls are significantly involved in complement activation, cytoskeleton organization and biogenesis, and regulation of cell growth and proliferation. Osmotic regulation is an essential homoeostatic process in all cells and tissues; however, disruption of ion and water transport is commonly noticed in pathologies such as cancer [58]. Alterations in membrane transporter function result in changes in cellular morphology and affect cellular behaviors such as attachment capability [59]. Moreover, membrane transporters are a principal agent in the transformation of cancer-associated cellular phenotypes and can promote cell proliferation, resistance to apoptosis, metabolic changes, angiogenesis, and migratory capabilities [59,60,61].

Finally, we analyzed the protein profiles of serum exosomes of NSCLC patients according to the metastatic spread of lung cancer cells (N feature of the pTNM staging) and observed that the frequency of detection of protein FAM166B positively correlates with tumor progression. Zhou et al. revealed that FAM166B could play a suppressive role in various cancers, such as breast cancer, head and neck cancer, and lung cancer [62]. Their study demonstrated a significantly lower FAM166B expression level at mRNA and protein levels in lung cancer compared to adjacent normal tissues. The role of FAM166B in lung cancer development has not been fully investigated yet. However, Zhou et al. revealed that in breast cancer, FAM166B is involved in the regulation of biological pathways closely related to glucose conversion, including glycolysis and the gluconeogenesis pathway. In addition, it plays a role in immune infiltration at the tumor site. Tumor cells are characterized by excessive consumption of glucose, and this metabolic reprogramming allows them to proliferate rapidly and also promotes tumor metastasis and therapy resistance [63]. Upregulation of FAM166B could reduce cellular metabolism and cell proliferation processes, and thus FAM166B can potentially inhibit cancer cell growth by affecting metabolic pathways. Furthermore, Zhou et al. reported that the expression level of FAM166B positively correlates with the immune infiltration level and may have a role in tumor immunity. Their analysis revealed that FAM166B may serve as a prognostic factor and indicate a better prognosis for breast cancer patients [62].

Our analysis revealed that the frequency of detection of olfactory receptor 52R1 and killer cell immunoglobulin-like receptor 2DL1 (KIR2DL1) negatively correlates with the spread of lung cancer cells in the lymph node. KIR2DL1 is an inhibitory receptor involved in the regulation of the function of natural killer cells (NK cells) through the recognition of major histocompatibility complex (MHC) class I molecules [64]. NK cells play a vital role in immunity to lung cancer and control of the metastatic spread of cancer via their ability to eliminate circulating tumor cells [65]. A study by He et al. showed that KIR2DL1 is expressed on tumor-infiltrating lymphocytes (TILs) and also on NSCLC cells, and its expression was observed in more NSCLC patients in advanced stages of disease (stages III and IV according to the AJCC) compared to patients with benign stages (stages I and II) [66]. Furthermore, they demonstrated that KIR2DL1 expression was correlated with poor prognosis in NSCLC patients. A study by Boyiadzis et al. revealed that activated NK cells produce large quantities of exosomes carrying various molecules, such as the activating NK cell receptor NKG2D, natural cytotoxicity receptors, perforin, granzyme B, transforming growth factor beta (TGF-β), and killer-cell immunoglobulin-like receptors (KIRs) [67]. These results suggest that NK cells are activated in the early stages of cancer and can remove KIRs via exosomes, but as the tumor progresses, NK cell activation is impaired and KIR secretion is inhibited.

Based on our own research and literature data, we demonstrated that many exosomal-derived proteins are involved in the various steps of NSCLC metastasis development, which is shown in Figure 9.

## 4. Materials and Methods

### 4.1. Patient Characteristics

The study cohort involved 15 NSCLC patients (*n* = 15). Of this number, there were 5 women and 10 men, aged 54 to 72 years (the mean age: 64.33 ± 5.55 years), who underwent lung resection (pulmonectomy or lobectomy) at the Department of Thoracic Surgery, General and Oncological Surgery at the Military Medical Academy Memorial Teaching Hospital of the Medical University of Lodz—Central Veterans’ Hospital, Lodz, Poland, during the years 2018–2019. Serum samples were obtained before surgery from all diagnosed NSCLC patients after excluding patients with no informed consent or preoperative neoadjuvant therapy. The patients were divided into groups depending on the postoperative histopathological evaluation of tumor tissue samples and classification according to the clinical staging system (AJCC, TNM) [68], which is shown in Table 1. Serum from 16 volunteers was obtained as a control.

### 4.2. Serum Collection

A total of 4 mL of venous blood was collected from all patients with NSCLC and the control group into tubes without anticoagulant and left at room temperature until clot formation (about 30–60 min). Next, the samples were centrifuged (1200× *g*, 10 min, 4 °C), and serum was separated and placed into new sterile tubes, frozen, and stored at −20 °C.

### 4.3. Isolation of Proteins from Serum Exosomes

The collected serum was thawed and centrifuged for 30 min at 4.4 rpm. 500 µL of the clear supernatant was transferred to new Eppendorf tubes, and 100 µL of Total Exosome Isolation Reagent (from serum) (Invitrogen™, Carlsbad, CA, USA) was added. The mixture was centrifuged until the serum became homogeneous and then incubated on ice for 30 min. Then, the samples were centrifuged for 10 min at 8.4 rpm at room temperature. The supernatant was discarded, and 200 µL of Exosome Resuspension Buffer from the Total Exosome RNA and Protein Isolation Kit (Invitrogen™, Carlsbad, NM, USA) was added to the exosome pellet and mixed thoroughly by pipetting the mixture several times. The analysis of exosomal antigen markers CD9, CD63, and CD81 was performed with the Human ProcartaPlex Mix and Match 3-plex Kit and the multiplex magnetic Luminex^®^ bead-based immunoassay using MAGPIX technology at the Research Laboratory CoreLab of the Medical University of Lodz (Appendix A).

### 4.4. LC/MS Analysis of Exosomal Proteins

The analysis of exosomal proteins was performed using the HPLC DIONEX UltiMate 3000 chromatograph equipped with a diode array detector (Thermo Fisher Scientific, Waltham, MA, USA) coupled with a micrOTOF-QIII mass spectrometer, equipped with ESI ionization, and a time-of-flight analyzer (Bruker Corporation, Billerica, MA, USA). All samples were separated by a reverse phase system using a 150 × 4.6 mm bioZen C8 HPLC column with a grain size of 3.6 µm (Phenomenex, Torrance, CA, USA). A mixture of water and acetonitrile with the addition of 0.1% formic acid was used as the mobile phase (all reagents were LC-MS grade). HPLC parameters were as follows: Gradient elution, component A—water, component B—acetonitrile, analysis program (B/A): 0–3 min 15/85, 3–5 min 30/70, 5–10 min 30/70, 10–50 min 40/60, 50–55 min 100/0, 55–60 min 100/0, 60–65 min 15/85, 65–70 min 15/85, analytical wavelengths of the DAD detector: 214, 220, 254, and 330 nm, injection volume 1 µL. For sample dilution, 10 µL of mixture was taken, and 300 µL of LC-MS water was added. Then, 270 µL of the obtained solution was introduced into a vial for analysis. MS spectrometer parameters were as follows: Drying gas flow—6.0 L/min, nebulizer temperature—50 °C, pressure—2.4 Bar, and capillary voltage—4000 V. The MS mass spectra were recorded in the positive ion mode. The recorded chromatograms were elaborated using DataAnalysis 4.2 software from Bruker. Peptide/protein identification was performed using the MASCOT database.

### 4.5. Hierarchical Clustering

Top proteins, exclusively detected in exosomes isolated from NSCLC and the control group, were used to create a bidirectional hierarchical clustering heatmap using their respective frequencies of occurrence. The hierarchical clustering method can be divided into several consecutive steps, starting with each of the n subjects (proteins or patient groups) forming its own cluster. In the first step, the two most similar subjects are joined to form one cluster (smallest branch); in the next step, the two most similar clusters are joined to form a bigger branch. The process is repeated until all clusters are defined [21]. This procedure results in a hierarchical dendrogram that highlights similarities and differences between the analyzed subjects. Calculation and visualization were performed using the Orange open-source machine learning and data visualization platform (https://orangedatamining.com/, accessed on 10 August 2023) as previously described by us [69].

### 4.6. Protein–Protein Interaction Network

Protein–protein interaction (PPI) networks of exosomal proteins exclusively observed in either the NSCLC patient cohort or control group were created and visualized using the STRING version 11.0 online platform (https://string-db.org/, accessed on 10 August 2023). Next, by using the Markov clustering (MCL) algorithm and the MCL inflation parameter = 3, protein interaction clusters were obtained. Functional enrichment analyses were (Lub: A functional enrichment analysis) performed using the STRING version 11.0 online platform and verified using the KEGG (Kyoto Encyclopedia of Genes and Genomes) PATHWAY Database (http://www.genome.ad.jp/kegg/, accessed on 10 August 2023) as described previously [69]. Additionally, a functional enrichment software tool, FunRich (v3.1.3), supported by the Vesiclepedia database (http://www.microvesicles.org/, accessed on 10 August 2023), was used to compare and analyze the cellular component, molecular functions, biological pathways, and biological processes differences associated with the proteins exclusively observed in exosomes from NSCLC patients vs. control group [70].

### 4.7. Analysis of Cancer-Associated Fibroblasts (CAFs) and M2 Macrophages Infiltration

An analysis of cancer-associated fibroblasts (CAFs) and M2 macrophage infiltration was performed using the Tumor Immune Estimation Resource—TIMER2.0 (http://timer.cistrome.org/, accessed on 10 August 2023) platform. TIMER2.0 utilizes immunedeconv, an R package that integrates six state-of-the-art algorithms (TIMER, xCell, MCP-counter, CIBERSORT, EPIC, and quanTIseq) to statistically predict tumor infiltration by selected immune cell types using the Cancer Genome Atlas (TCGA) database. The data were analyzed and visualized using the xCell algorithm [71,72,73].

## 5. Conclusions

Our study revealed a significant difference in the protein profile of serum exosomes derived from NSCLC compared to the control group and indicated several proteins that may be used as a potential diagnostic marker for NSCLC patients. Identified exosomal proteins may have a significant impact on osmoregulation as well as cell adhesion, motility, and differentiation, contributing to the development of NSCLC cancer. We observed changes in the frequency of detection of the following proteins: FAM166B, killer cell immunoglobulin-like receptor 2DL1, and olfactory receptor 52R1 in serum exosomes according to the presence of metastasis to lymph nodes. These results suggest that these proteins could play a significant role in disease progression. They may be a potential non-invasive diagnostic marker with a negative prognostic value in liquid biopsy. However, our observation requires further investigation. We demonstrated that interleukin-34, HLA class II histocompatibility antigen, DM alpha chain, and HLA class II histocompatibility antigen, DO beta chain, have a significant impact on CAF and TAM infiltration, and the presence of these proteins in serum exosomes derived from NSCLC patients may indicate poor prognosis for NSCLC patients and result in failure of their therapy.

## Figures and Tables

**Figure 1 ijms-24-13669-f001:**
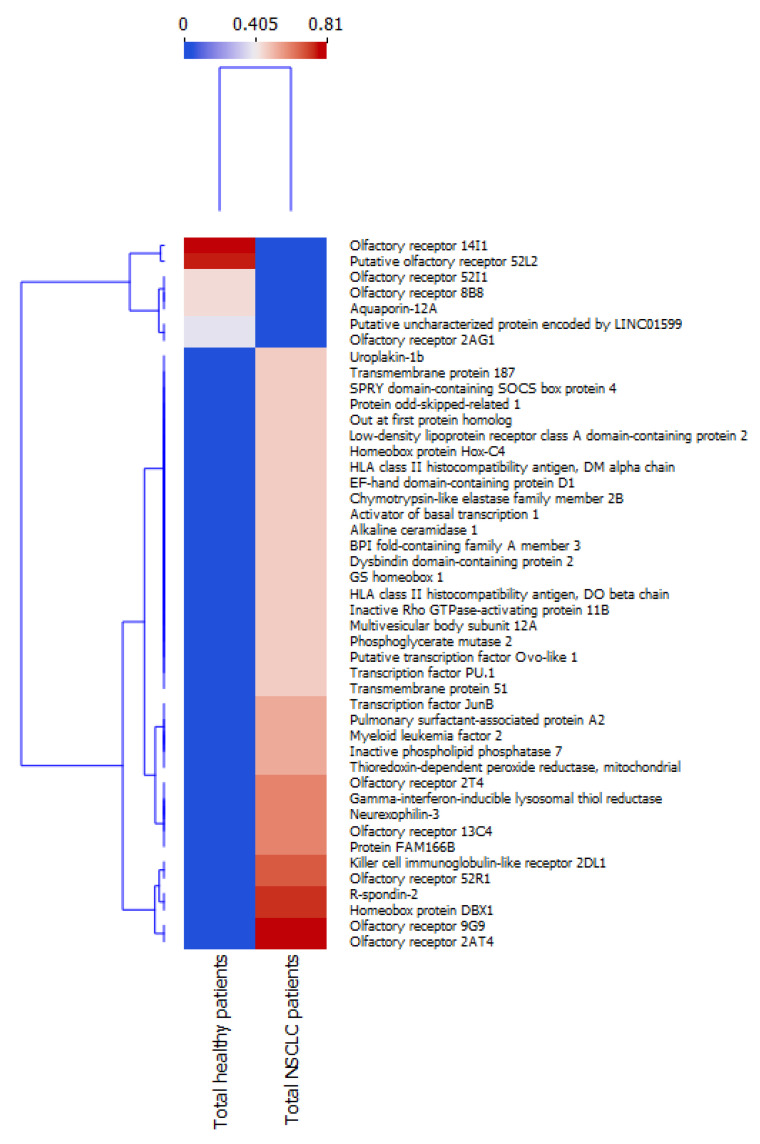
Hierarchical clustering analysis of NSCLC and healthy patients.

**Figure 2 ijms-24-13669-f002:**
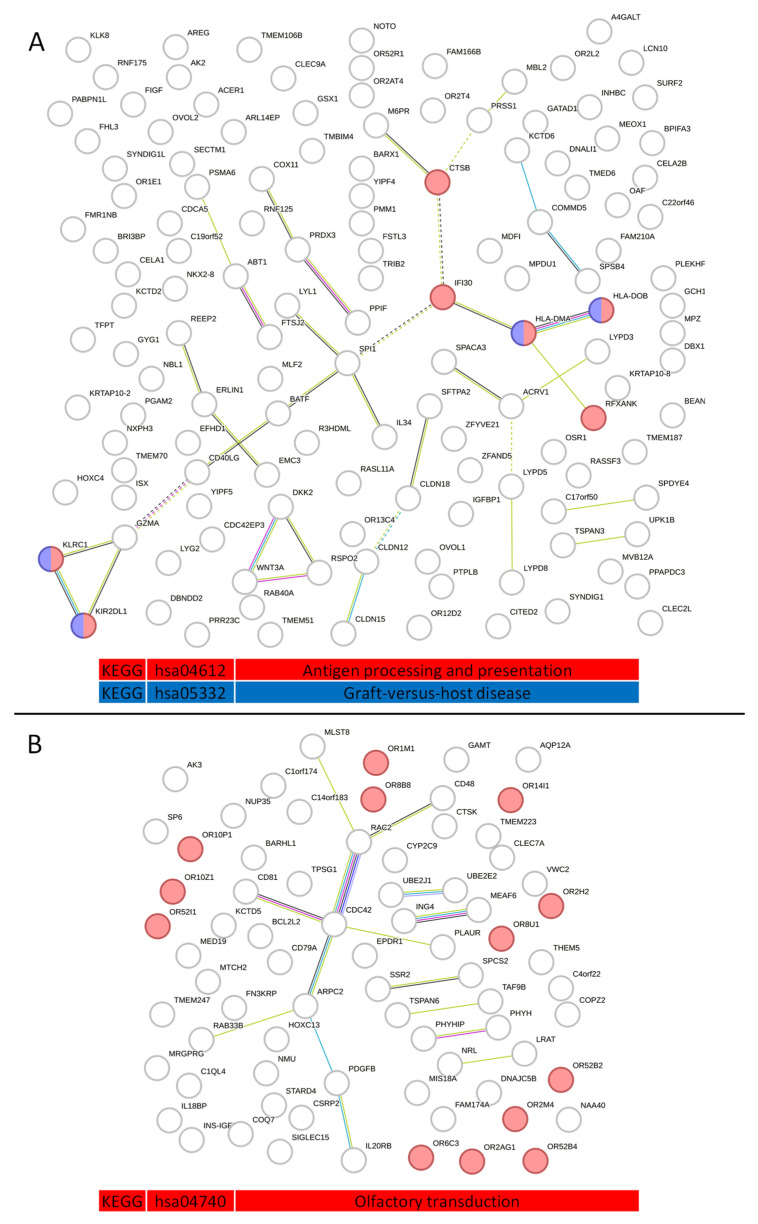
NSCLC (**A**) and control (**B**) group serum exosome-derived protein–protein interaction network and functional enrichments.

**Figure 3 ijms-24-13669-f003:**
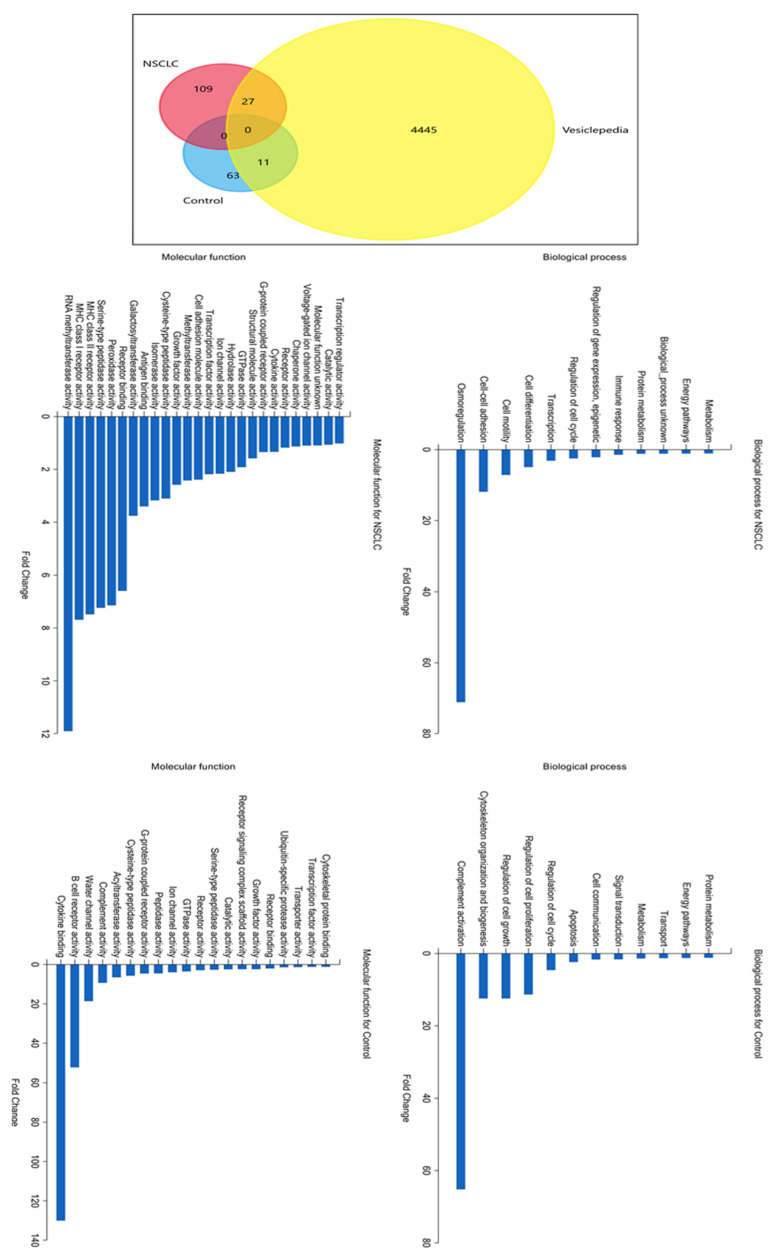
Functional enrichment of NSCLC and control group-derived exosome proteins.

**Figure 4 ijms-24-13669-f004:**
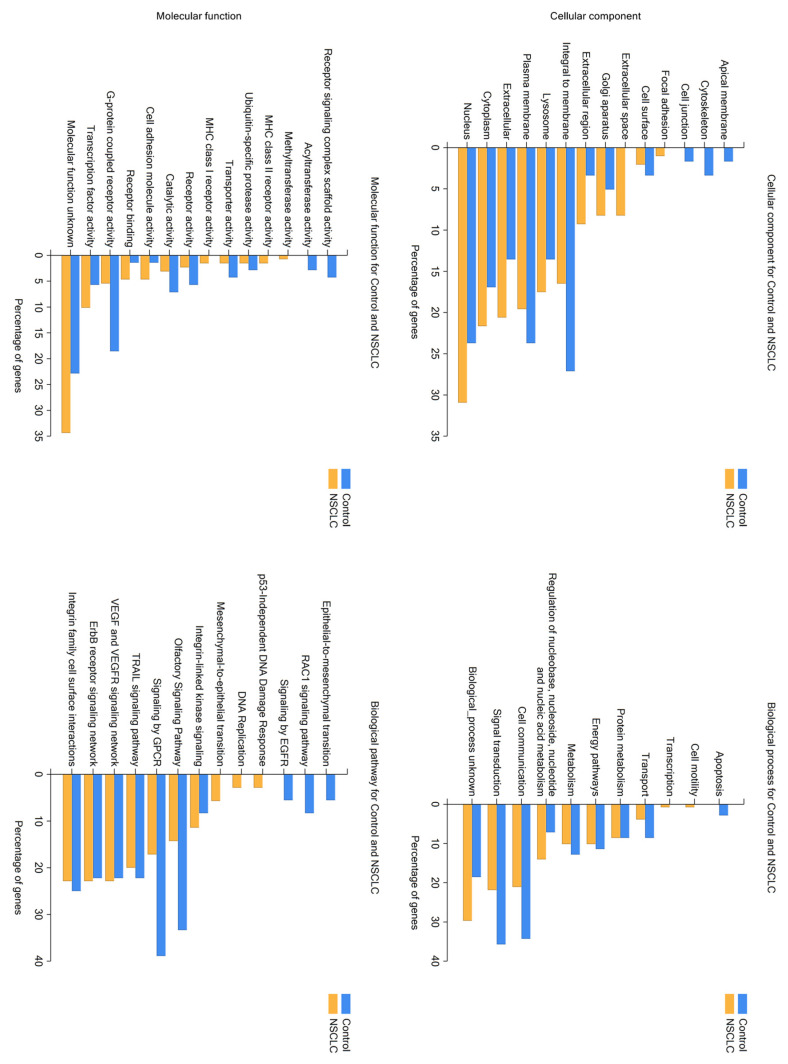
Comparative analysis of protein functional enrichment of NSCLC and control group-derived exosome proteins.

**Figure 5 ijms-24-13669-f005:**
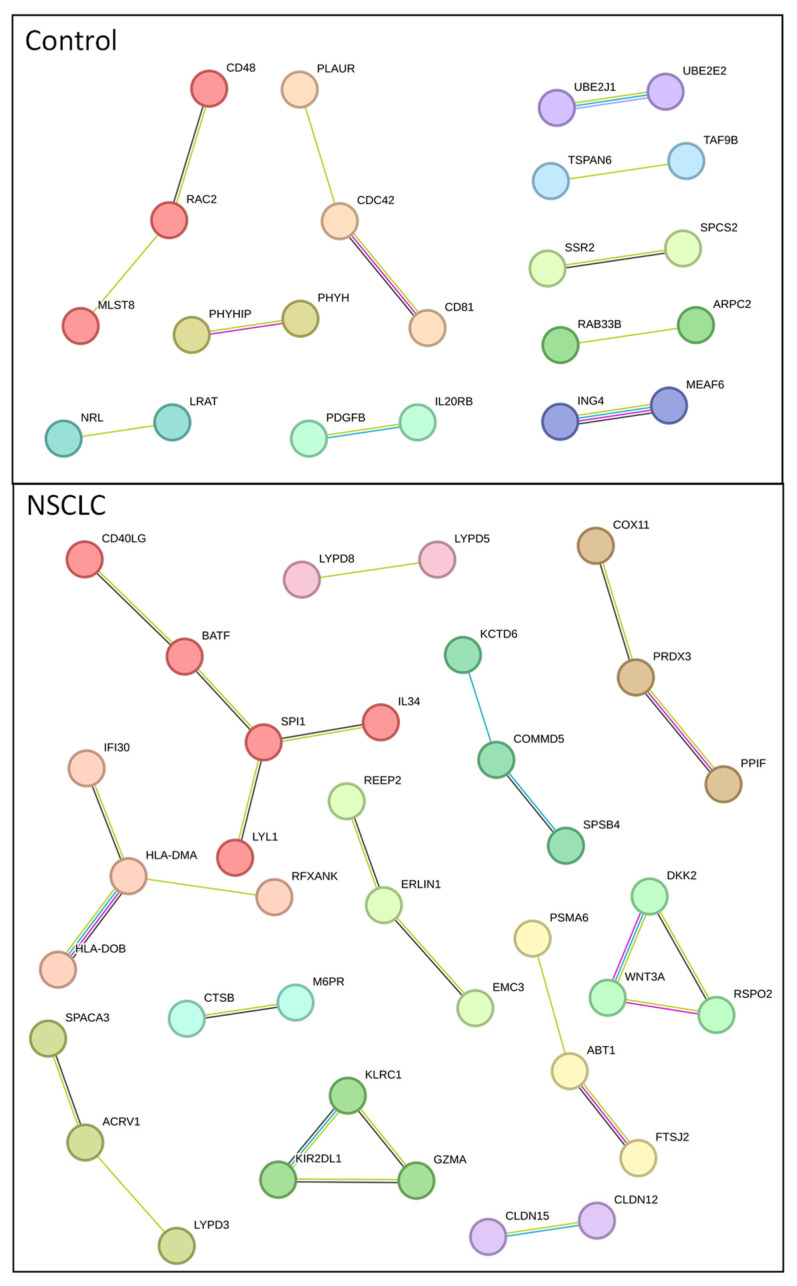
Control and NSCLC serum exosome-derived protein hub clusters.

**Figure 6 ijms-24-13669-f006:**
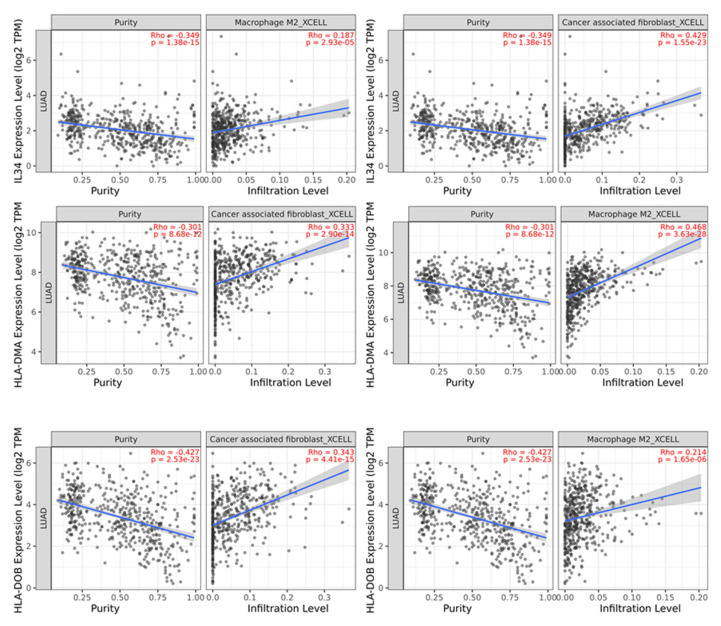
Involvement of chosen exosome proteins exclusive to NSCLC patients with cancer-associated fibroblasts and M2 macrophage infiltration.

**Figure 7 ijms-24-13669-f007:**
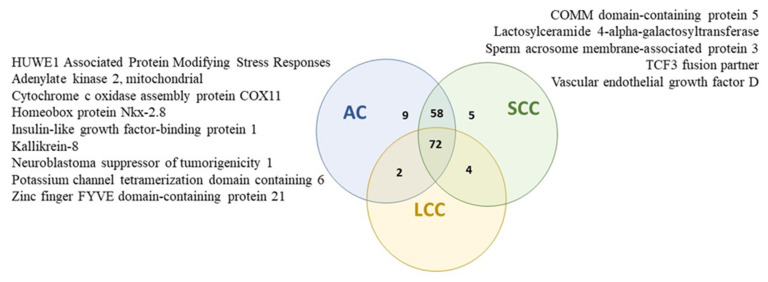
Exosomal proteins detected exclusively in patients with diagnosed AC and SCC subtypes.

**Figure 8 ijms-24-13669-f008:**
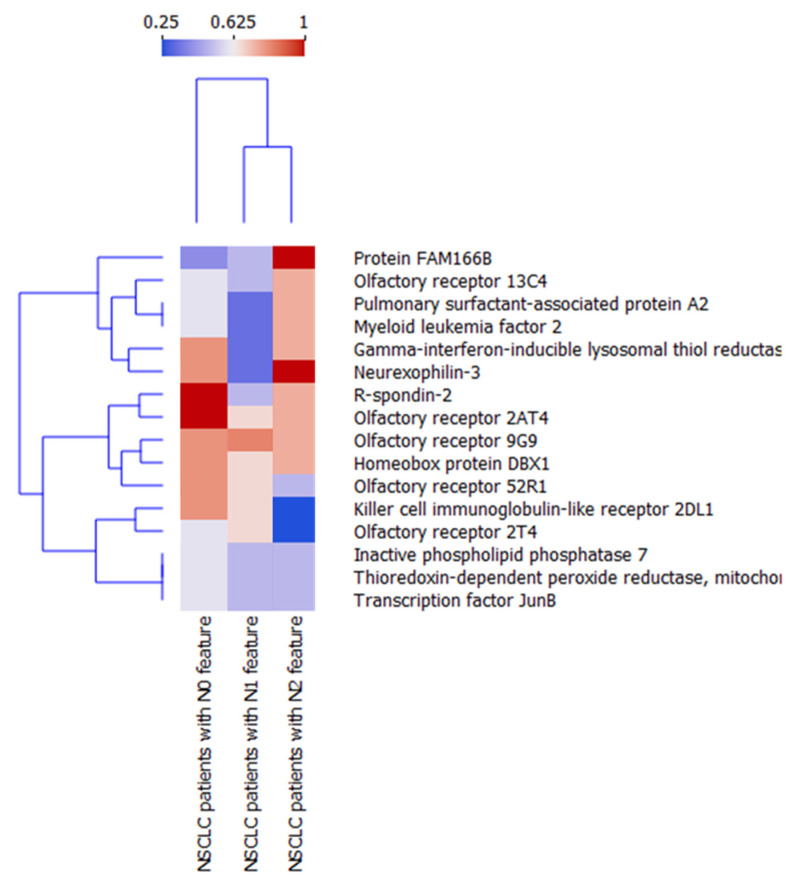
Hierarchical clustering analysis of NSCLC patients N feature subgroups.

**Figure 9 ijms-24-13669-f009:**
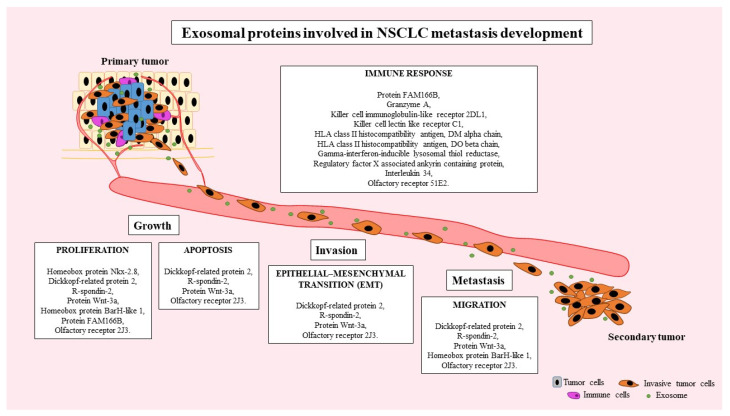
Involvement of exosomal-derived proteins in various steps of NSCLC metastasis development.

**Table 1 ijms-24-13669-t001:** Histopathological characteristics of tumor tissue samples and classification according to clinical staging system (AJCC, TNM).

Tumor Features	Number of Cases (*n*)	Total Percentage of Cases
Histopathological subtype of NSCLC		
SCC ^a^	6	40%
AC ^b^	8	53%
LCC ^c^	1	7%
Stage of cancer development		
according to AJCC ^d^ staging system		
Stage I	3	20%
Stage II	6	40%
Stage III	6	40%
Presence of lymph node metastasis		
according to the pTNM ^e^ staging system		
N0	5	33%
N1	6	40%
N2	4	27%
Tumor size according to the pTNM staging system		
T1a + T1b	6	40%
T2a + T2b	5	33%
T3 + T4	4	27%

^a^ SCC—squamous cell carcinoma; ^b^ AC—adenocarcinoma; ^c^ LCC—large-cell carcinoma; ^d^ AJCC—American Joint Committee on Cancer Staging according to the IASCLC Staging Project 7th ed. (2010) Cancer; ^e^ pTNM—post-operative tumor node metastasis staging system according to the WHO Histological Typing of Lung Tumor.

## Data Availability

The data used to support the findings of this study are available from the corresponding author upon request.

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
