# Peer review of "The Comparison of Serum Exosome Protein Profile in Diagnosis of NSCLC Patients"

_ijms, 2023, doi:10.3390/ijms241813669_

Round 1
Reviewer 1 Report
I thank the authors for submitting the manuscript to the journal.
The paper reports the serum exosomal analysis of NSCLC patients to identify protein biomarkers having consequential implications in the etiology and disease progression of NSCLC. The methods followed by the authors are standard and authors have provided a discussion on the biomarkers as well. My only suggestion to the authors would be to include a diagram showcasing the markers and the associated pathways modulating the progression of the disease, provided that sufficient literature data is available for support.
The quality of language and grammar is fine.
Author Response
The paper reports the serum exosomal analysis of NSCLC patients to identify protein biomarkers having consequential implications in the etiology and disease progression of NSCLC. The methods followed by the authors are standard and authors have provided a discussion on the biomarkers as well. My only suggestion to the authors would be to include a diagram show casing the markers and the associated pathways modulating the progression of the disease, provided that sufficient literature data is available for support.
Thanks for Reviewer’s suggestion, however, due to the fact that the exact mechanism of action of some of the identified exosomal proteins is not yet known in this type of cancer, we cannot make a proper detailed diagram, that would accurately represent the biochemical/signaling pathway relationships.
We have proposed a division of markers according to their known molecular and biological functions (Table S1). Therefore, we propose to include in the article a general figure representing the importance of the biological processes in which these protein markers are involved in the course of NSCLC metastasis development (Figure 9).
Reviewer 2 Report
The authors present an interesting study in which liquid biopsies obtained from non-small cell lung cancer and a corresponding control group, and were then processed to isolate the exosomes from each in order to screen them and determine the correlation in their cargo and cancer progression. This study identified 150 NSCLC-associated proteins, with three in particular shown to demonstrate a significant influence on cancer progression. In short, this study highlights the potential of screening exosomes as a means of determining clinical risks of relevant diseases, but also identifies key molecules in lung cancer with potential rom the perspective of therapeutic intervention.
In reviewing the manuscript I made a number of observations. The following should be considered when preparing a suitable revision.
1. It would be useful if the abbreviation of NSCLC was expanded upon in the abstract such that it was clearer to prospective readers as to what the clinical cohort of the piece is
2. Does freezing the serum affect the exosome population?
3. When the authors say the serum was pooled, does this imply that all samples within each clinical cohort were polled together for the analysis or what is meant by this?
4. Were any sizing data obtained or the exosomes to determine they fell into the defined ranges?
5. It is relatively difficult to read the data contained in the figures given the resolution and sizing of the data – the authors should review their presentation of the data and improve on this where possible
Author Response
- It would be useful if the abbreviation of NSCLC was expanded upon in the abstract such that it was clearer to prospective readers as to what the clinical cohort of the piece is
As suggested the Reviewer, we extended the indicated abbreviation.
- Does freezing the serum affect the exosome population?
The study of Gelibter et al. demonstrated that plasma storage at -80°C leads to a time‐dependent decrease in the concentration of extracellular vesicles (EVs)[1]. However, short-time storage (few weeks storage) did not contribute to significant reduction of EVs concentration. A significant reduction in EVs load occurred during long-time storage (after 6 months of plasma storage).
[1] S. Gelibteret al., ‘The impact of storage on extracellular vesicles: A systematic study’, J Extracell Vesicles, vol. 11, no. 2, p. e12162, Feb. 2022, doi: 10.1002/jev2.12162.
- When the authors say the serum was pooled, does this imply that all samples within each clinical cohort were polled together for the analysis or what is meant by this?
Unfortunately, there was a misunderstanding, because instead of the word "pooled", it would be more appropriate to use the word "collected". Each patient sample was assayed separately.
- Were any sizing data obtained or the exosomes to determine they fell into the defined ranges?
In our study, we did not measure the size to identify exosomes but used the analysis of exosomal marker proteins.
- It is relatively difficult to read the data contained in the figures given the resolution and sizing of the data – the authors should review their presentation of the data and improve on this where possible
We have made changes where possible.
Reviewer 3 Report
The authors performed analysis of the exosomal proteins content from the liquid biopsy of the serum of 15 non-small cell lung cancer (NSCLC) patients and 16 donors without NSCLC. They determined several proteins specific for
NSCLC group and performed bioiiformatic analysis (hierarchical clustering, protein-protein interaction, analysis of protein enrichment in the molecular functions and
biological processes, analysis of protein hub clusters, etc) deciphering their functional role and diagnostic/prognostic significance. They also attempted to stratify the exosomal protein profiles of NSCLC patients according to the histopathological sub-type of tumor and tumor size, as welll as the state of regional lymph nodes. As for the clinical value of the findings, the data presented in this manuscript suggests that analysis of exosomal proteome in the serum of NSCLC patients may be used as a prognostic factor to predict immunotherapy outcome. The authors also identified several proteins within exosomes that correlate with tumor progression, etc.
Please find below my comments and suggestions.
Please define all abbreviations upon first use, both in the abstract and throughout the text
Line 45. “Liquid biopsy represents a novel and non-invasive approach to test tumor biomarkers in biological fluids, such as blood, saliva, urine, cerebrospinal fluid.” - and Bronchoalveolar Lavage (BAL), of course, in case of lungs!
Line 48. “materials secreted by tumors into the circulation, such as circulating tumor cells (CTCs)”. I would not call the circulating tumor cells “materials secreted by tumors”. Please change the wording.
Line 204. “Heaving established” - having established
The weakness of the study is the small size of the study cohort.
Author Response
Please define all abbreviations upon first use, both in the abstract and throughout the text
We have introduced abbreviations upon first use.
Line 45. “Liquid biopsy represents a novel and non-invasive approach to test tumor biomarkers in biological fluids, such as blood, saliva, urine, cerebrospinal fluid.” - and Bronchoalveolar Lavage (BAL), of course, in case of lungs!
We have added to the article this very important material used in liquid biopsy of lung cancer patients.
Line 48. “materials secreted by tumors into the circulation, such as circulating tumor cells (CTCs)”. I would not call the circulating tumor cells “materials secreted by tumors”. Please change the wording.
We've made changes: ‘This method involves an analysis of circulating tumor cells (CTCs) or materials secreted by tumors into the circulation, such as cell-free nucleic acids (cfNAs) and extracellular vesicles (EVs).’
Line 204. “Heaving established” - having established
We have made this change.
The weakness of the study is the small size of the study cohort.
We thank the Reviewer for comments. Indeed, our study concerned a small group of patients, which may be associated with a bias in statistics - this is a limitation of our work. Despite the fact that the study was single-center, it concerned the Polish population, which is innovative.
Round 2
Reviewer 1 Report
I thank the authors for addressing my comments and revising the manuscript.
The quality of English language appears to be fine.
Reviewer 2 Report
The authors have addressed my comments.